# Alveolar Volume Following Different Timings of Secondary Bone Grafting in Patients with Unilateral Cleft Lip and Palate. A Pilot Study

**DOI:** 10.3390/jcm10163524

**Published:** 2021-08-11

**Authors:** Andrzej Brudnicki, Piotr A. Regulski, Ewa Sawicka, Piotr S. Fudalej

**Affiliations:** 1Department of Maxillofacial Surgery, Clinic of Pediatric Surgery, Institute of Mother and Child, 01-211 Warsaw, Poland; e.sawicka@chello.pl; 2Dentomaxillofacial Radiology Department, Medical University of Warsaw, 02-091 Warsaw, Poland; pregulski@interia.pl; 3Center of Digital Science and Technology, Cardinal Stefan Wyszynski University, 01-815 Warsaw, Poland; 4Department of Orthodontics and Dentofacial Orthopedics, School of Dental Medicine, University of Bern, 3012 Bern, Switzerland; pfudalej@gmail.com; 5Department of Orthodontics, Institute of Dentistry and Oral Sciences, Palacký University Olomouc, 779 00 Olomouc, Czech Republic; 6Department of Orthodontics, Jagiellonian University, 31-155 Krakow, Poland

**Keywords:** alveolar bone grafting, UCLP, alveolar volume, three-dimensional outcome of bone surgery, CT measurement of alveolar volume

## Abstract

This study was intended to evaluate the relationship between secondary alveolar bone grafting (SABG) timing and the alveolar volume in patients with unilateral cleft lip and palate (UCLP). The material consisted of CTs of 35 patients (17 males, 18 females) with UCLP who underwent a one-stage primary cleft repair at a mean age of 8.4 months and SABG at different timings ranges of 1.8–18.8 years. The mean age at CT was 17.2 years. The relative coefficient (*Ꞷ*) which was independent from factors such as individual maxillary size, gender or age at the CT was introduced in order to compare volumes of the cleft-side in relation to the non-cleft-side alveolus. Pearson correlation coefficient *r* between *Ꞷ* coefficient and SABG timing was weak negative (*r* = −0.34, *p* = 0.045). The multiple regression analysis implied that the dependent variable-*Ꞷ* coefficient was associated with independent variables (cleft repair and SABG timings and age at CT) with *r*^2^ = 0.228. Only patient’s age at SABG explained the dependent variable (*p* = 0.003). The study cautiously indicates a tendency to larger alveolar volume following earlier timing of SABG. Nevertheless, the further research on a larger group of patients should be performed before formulating any clinical indications.

## 1. Introduction

Alveolar bone grafting—the surgical procedure aiming to restore continuity of alveolus—has become obligatory in the contemporary management of alveolar cleft defects and its clinical benefits have been widely reported in the literature [1]. The procedure performed at the time of lip closure was called primary bone grafting while the procedure performed thereafter as the next step in the surgical protocol was called secondary bone grafting [2]. The postponement of the secondary alveolar bone grafting (SABG) until 9–11 years of age [3] has become the standard of treatment in majority of cleft centers worldwide [4]. The main reason for this postponement was the fear of potentially more significant interference with maxillary growth at younger age and subsequent maxillary restriction [5,6,7]. These assumptions were questioned by the later studies, partly based on long-term observations of matured patients—they implied that SABG had comparable influence on subsequent maxillary growth regardless of its timing [8,9,10,11]. Furthermore, it has been proved that SABG at an earlier age is associated with less donor-site morbidity and definitely does not increase donor site symptoms, surgical duration or hospitalization following surgery [12].

In line with this, one can observe the recent tendency to perform SABG earlier than previously recommended [8,13,14,15,16,17]. Nevertheless, a definitive conclusion cannot be reached [18] and the relationship between the timing of SABG and its long-term outcome is still debated. One of the reasons of ongoing debate is the paucity of studies looking at the effects of SABG performed before 5 years of age. To the best of our knowledge, our older report using two-dimensional imaging and investigating the outcome of SABG in patients between 2 and 18 years is probably the only exception [16]. Meanwhile, new three-dimensional methods seem to have become the state of the art in the contemporary assessments of bone surgery outcome [19,20], because they provide precise and accurate representations of the anatomical structures and pathological processes [4], while two-dimensional methods have some limitations such as no volumetric information, enlargement, distortion, and overlap of anatomical structures, and limitations for anatomical landmarks identification, thus affecting the accurate measurements [21].

Hence, the aim of this study was to assess the alveolar bone volume in patients with unilateral cleft lip and palate (UCLP) after SABG performed at different timings (including age of much younger patients than 5 years at the procedure). The research hypothesis of this pilot study was that SABG performed in younger patients results in larger alveolar volume.

## 2. Materials and Methods

The present study is a part of a larger research project which was approved by institutional ethics committee (The Bioethics Committee of the Institute of Mother and Child (IMC), reference 21/2013). All subjects consented in writing to participate in this study. The surgeons who carried out primary cleft repairs and SABGs were not involved in this study.

### 2.1. Subjects

This is a retrospective evaluation of a cohort of patients born with non-syndromic complete UCLP and treated surgically in a single institution using the same surgical techniques.

The medical records of 94 patients with UCLP treated at IMC who had a multi-slice spiral computed tomography (CT) scan of maxillofacial region performed from August 2012 to July 2019 were reviewed. The patients who met the following inclusion criteria were selected: (1) non-syndromic UCLP, (2) after SABG, (3) all surgical repairs performed exclusively in our institution, (4) complete medical record including CT scan performed >2 years after SABG. The patients in whom CT scans were taken before SABG were excluded from further evaluation. The presence of Simonarts’ band was not an inclusion/exclusion criterion.

### 2.2. Surgical Treatment

All patients underwent a one-stage primary cleft repair (soft and hard palate as well as lip repair done simultaneously) elaborated recently in literature [22] and SABG procedure according to principles described by Boyne and Sands [3]. In short, SABG included the separation of the attached layers of the oral and nasal mucous membrane covering the cleft in the alveolar region in aim to create a space for the bone graft. The elevation of palatal periosteum was minimized. The cancellous bone was harvested from the anterior part of the iliac crest in all cases, using the same surgical technique regardless the patient’s age [12]. Shortly after harvesting, the portion of cancellous bone was always fixed firmly between the alveolar bone edges of the cleft and finally covered by the vestibular gingival flaps.

### 2.3. Computed Tomography (CT) Evaluation

The postoperative CT images were acquired with spiral computer tomography (Brilliance 64, Philips, Amsterdam, The Netherlands). Exposure parameters were adjusted to assess bone structures and were as follows: tube voltage: 100–120 kVP, current: 78–180 mAs, acquired slice thickness: 0.3 mm, spacing between slices: 0.45 mm, voxel size: 0.3 × 0.3 × 0.9 mm, slice resolution: 512 × 512 voxels.

The CT files of the identified patients were retrieved from their medical records and analyzed. Segmentation and measurements were performed using the VisNow-Plugin-Medical library—a Java plugin for the open-source VisNow platform that allows medical analysis and visualization. The platform is available at https://gitlab.com/cnt-uksw/visualization/VisNow [23]. An investigator who analyzed CT scans was blinded regarding treatment details of participants. The alveolar grafted bone area was traced on the horizontal CT sections build of interpolation of three CT slices and of a total thickness of 0.9 mm, using the drawing tools of image-data processing software. The images were enlarged manually. The labial and palatal outline of bone localized between the upper central incisor and canine (between the lines of the widest diameter of the bordering roots) were followed, excluding root of the lateral incisor if present (Figure 1). In the vertical dimension, the tracings were performed from the level of the first visible sign of bone till the level of root apex of the upper central incisor. The outline of bone on each CT slice was plotted and the volume of each measured CT segment was automatically calculated by multiplication of the surface area and its thickness. A 3D reconstruction of the whole bone area results from the sum of all measured single volumes of CT segments. In order to calibrate the individual outcome of existing alveolar bone volume, specific for particular patient, the bone volume on the contralateral non-cleft side of alveolus was measured according to the same way for comparison with that of the cleft side. The values of the alveolar bone volume were calculated and assigned to the corresponding patients.

### 2.4. Statistical Analysis

For all statistical calculations, Statistica 13 software (StatSoft, Tulsa, OK, USA) was used. The difference between cleft and non-cleft sides was confirmed with paired Student’s *t*-test per patient.

The relative coefficient (*Ꞷ*) which was independent from factors such as individual maxillary size, gender or age at the moment of CT scan and was introduced in order to compare volumes of the cleft side in relation to the non-cleft side:(1)$$Ꞷ=\frac{V_{cleft-side}}{V_{cleft-side}+V_{non-cleft-side}}$$
where *V_cleft-side_* and *V_non-cleft-side_* are the volumes of segmented alveolar bone on cleft and non-cleft sites, respectively.

Correlation analysis between *Ꞷ* coefficient and the patient’s age at the primary cleft repair surgery and SABG was performed with Pearson correlation coefficient (*r*). The relationship based on *r* value within a range from −1.0 to −0.7 was interpreted as a strong negative relationship, −0.7 to −0.5—as moderate negative, −0.5 to −0.3—as weak negative, −0.3 to 0.3 as no relationship, 0.3 to 0.5 as weak positive, 0.5 to 0.7 as moderate positive, and 0.7 to 1.0 as strong positive. The correlation coefficient was also calculated between the age at CT scan and *Ꞷ* coefficient in order to evaluate the potential influence of this factor on the final result. The correlation significance was tested with Student’s *t*-test.

The effect of surgeon performing SABG on *Ꞷ* coefficient was assessed with the Kruskal–Wallis test, while the effect of presence/absence of lateral incisor on *Ꞷ* coefficient was assessed with the Mann–Whitney U test.

The multiple regression analysis with *Ꞷ* coefficient as dependent variable and age at cleft repair, age at SABG, age at CT.

The power analysis was performed in order to estimate the sample size needed for future prospective study with desired power equal to 0.8.

### 2.5. Error of the Method

Before the beginning of the study an appropriate standard for measurement was worked out and then adopted. In order to assess intra-rater reliability the differences between the first and second measurements on 7 (20%) randomly selected CT files were determined by the same rater one month later. Intra-rater reliability was assessed using interclass correlation coefficients (ICC) [24] to assess the degree that rater provided of consistency in his ratings across subjects. The qualitative rate of agreement based on ICC values with reliability was poor for values less than 0.40, fair for values between 0.40 and 0.59, good for values between 0.60 and 0.79, and excellent for values between 0.80 and 1.0.

## 3. Results

Finally, the sample comprised 35 patients (17 males, 18 females) born between July 1993 and May 2002 who had one-stage primary repair of complete UCLP at a mean age of 8.4 months and SABG at a mean age of 7.6 years (Table 1). None of them had Simonart’s band.

The SABG was performed by 3 operators from the cleft team of the same institution. CT scans were taken at least 1.8 years after SABG (median 9.9, IQR = 8.0).

On the cleft side, the lateral incisor erupted in 19 (54.3%) subjects, while the canine erupted in all subjects. On the non-cleft side both lateral incisor and canine erupted in all subjects.

The analysis of method error demonstrated high reliability of the method. The ICC = 0.999 suggested a minimal amount of measurement error introduced during assessments.

The average measurements of alveolar bone volume on the cleft-side, non-cleft side, difference between them and *Ꞷ* coefficient were: 417.0 ± 177.9 mm^3^, 781.2 ± 220.4 mm^3^, 364.3 ± 217.9 mm^3^, 0.34 ± 0.09, respectively. The difference between the cleft-side and non-cleft-side volumes was statistically significant (*p* < 0.0001)—Figure 2.

Correlation analysis between *Ꞷ* coefficient and patient’s age at SABG showed a weak negative relationship (*r* = −0.34, *p* = 0.045). There was no significant correlation between *Ꞷ* coefficient and patient’s age at the primary cleft repair (*r* = −0.24, *p* = 0.16) or *Ꞷ* coefficient and age at CT scan (*r* = 0.29, *p* = 0.08)—Figure 3, Table 2.

The effect of surgeon performing SABG on *Ꞷ* coefficient was not significant (*p* = 0.48) and the effect of presence/absence of lateral incisor on *Ꞷ* coefficient was also not significant (*p* = 0.228). Therefore both variables were not included in multiple regression analysis. The multiple regression analysis revealed that the model was not statistically significant. The dependent variable—*Ꞷ* coefficient was associated with independent variables (age at cleft repair, age at SABG, and age at CT) with *r* = 0.47, *r*^2^ = 0.22. The independent variable that was significantly predicting the *Ꞷ* coefficient was patient’s age at SABG (*p* = 0.003) and the regression coefficient = −0.008 meant that a 1 year delay in performing SABG was associated with a decrease of the *Ꞷ* coefficient by 0.008 (Table 3). Taking into account the obtained result from the multiple regression and the Equation (1), the earlier timing of SABG can provide about 9.4 mm^3^ of new bone yearly.

The power analysis revealed that a sufficient number of cases required for further prospective research should be 51 at least. The sample size should be increased by the potential dropout of patients to follow-up during clinical trial. If we assume that about 20% of patient’s are lost to follow-up, then 62 patients in total are required for the trial.

## 4. Discussion

It has been suggested that the timing of the SABG is a critical variable affecting its outcome [24,25,26,27,28]. Nonetheless, surprisingly few studies evaluated results of SABG performed earlier than a widely accepted age range of 8–11 years that was assumed to be optimal for clinical success since the 1970s [3,5]. In particular, SABG performed during the deciduous dentition stage (1–6 years of age) has not been well researched. Therefore, the aim of this pilot study was to assess the volume of the bone grafted at various ages (1.8 to 18.8 years) in patients with UCLP.

The present study indicates a tendency to a larger alveolar volume on the cleft side in patients who had SABG at younger age. Although correlation coefficients seem low and might be regarded as clinically unimportant, our results imply that early timing of SABG could be a viable therapeutic option provided comparable outcome will be found in a larger sample. In a biological perspective, current results demonstrate favorable bone integration and regeneration following SABG in patients as young as 2–5 years of age. It can be assumed that registered volumetric measurements are related to regeneration capacity of bone tissue. Studies in the field of regenerative medicine indicate that mesenchymal stem cells (MSCs) are good candidates for various clinical applications including bone reconstruction. This is because MSCs have the potential to differentiate into multiple cell types, including osteoblasts. Contemporary studies of MSCs, although mainly focused on aging, suggest an age-related tendency to decline in regenerative capacity [29,30,31].

The present study showed that alveolar bone volume was significantly less on the cleft side (*p* < 0.001) in comparison to the non-cleft side, which is in agreement with previous long-term volumetric observations indicating significantly less bone on the grafted cleft side (*p* < 0.001), despite satisfactory functional results [32]. It can be assumed that the alveolar bone volume observed in this study should be attributed not only to the late results of bone grafting surgery but also its growth during the developmental period, since it has already been well documented that grafted bone volume decreases by about 50% in the first year after surgery, and then remains almost constant in the following 2 years [33]. In a long-term perspective, the volume of reconstructed alveolar bone, once integrated, grows in a similar fashion as in the contralateral non-cleft side (Figure 3)—*Ꞷ* coefficient and age at CT scan were not correlated (*r* = 0.29, *p* = 0.08), which means that the relation of bone volume of the cleft side and non-cleft side stays almost constant during the investigated period of developmental age. It is in contradiction to what was previously assumed that ‘*inserted bone does become shorter over the years*’ and ‘*The bone furthermore does not show any apposition at the frontal site, according to influences on the basis of functional stimuli in normal frontal bite of the deciduous incisors. The frontal ends of the alveolar process conjugated by bone are rather retarding in their development in all three dimensions. This reminds us of the old experiences that a too early touch of the bony cleft is contrary to growth.*’ These fears and assumptions were the ground for the conclusion that ‘*the practice of primary and early secondary bone grafting of the alveolar cleft should be abandoned*’ [34].

The study by Brudnicki et al. [16] based on 2D radiographic documentation of 108 patients with UCLP revealed the correlation between the age of a patient during SABG and its outcome. However, the statistical significance was registered only with respect to the group of patients aged 6–16 years and not confirmed in the group of patients younger than 6 years at the time of bone grafting. It suggests a non-linear character of the registered tendency. Hence, the studies aimed to verify these findings should be based on numerous and homogenous material including that from very young patients.

The tendency observed in the present study is weak and variability of the small study group precludes more precise evaluation of the registered correlation and its character at this point. That could also explain why some previous studies based on 3D assessment failed to indicate any significant influence of the procedure timing on the outcome of SABG [35,36]. Presumably, the narrow range of the evaluated SABG timing (between 8 and 12 years of age) and small number of participants in these studies could have made the registration of any subtle correlations, at a statistically significant level, very difficult.

The segmentation and analysis process were performed with the specially designed method for this research module implemented on VisNow-Medical Plugin platform. The module allowed for manual contour tracing of the alveolar bone on cleft and non-cleft sides separately. The tracing was performed on horizontal CT sections which were connected together to obtain a volume. The total process of segmentation in this pilot study was time-consuming, therefore, further research will be focused on the automatization of the alveolar bone tracing. However, some semi-automatic tracing of alveolar bone or teeth has already been reported [37,38,39].

It should be underlined that this study aimed to answer questions pertaining to the volume of alveolar bone after its reconstruction, which is only one of several aspects determining the clinical success of SABG. Other outcome variables that could be equally important when analyzing the clinical success of SABG (e.g., residual oronasal communication, eruption of lateral incisor or canine into cleft site, ability to restore missing tooth in cleft with implant or orthodontic substitution, periodontal defect, degree of maxillary hypoplasia) are not accounted for when interpreting the result of this study.

The limitations of the present study include its retrospective character, a relatively small number of participants and other possible biases such as missing details on orthodontic expansion prior to bone grafting. It cannot be excluded that wear of expansion plates affected the amount of grafted bone volume long-term and the effect was different relative to the age of bone grafting: 2- to 4-year-olds were unlikely to wear any expansion plates, while 5- to 6-year-olds (or older) might have worn them. Furthermore, 3D radiographic scans were not obligatory in the management of patients with UCLP at that time, hence they were not performed in all consecutive patients.

The results of this pilot study imply that further investigations with larger samples are warranted to obtain more conclusive results. A larger group should be also considered in further studies in order to evaluate the linearity of correlation between cleft side volume or its coefficient and age at SABG before formulating any clinical indications. The low doses CBCT as routine in assessment of UCLP patients [40,41] would contribute to collecting more informative material for further studies.

## 5. Conclusions

Within the limitations of this study (i.e., relatively small sample size and significant range of timing of bone grafting) we cautiously conclude that our research hypothesis was partially confirmed: alveolar bone volume on the cleft side seemed not to be less when SABG was performed in younger patients in comparison to SABG performed in older patients. However, this preliminary conclusion should be confirmed on a larger sample of subjects with similar diagnosis and management to formulate robust clinical indications.

## Figures and Tables

**Figure 1 jcm-10-03524-f001:**
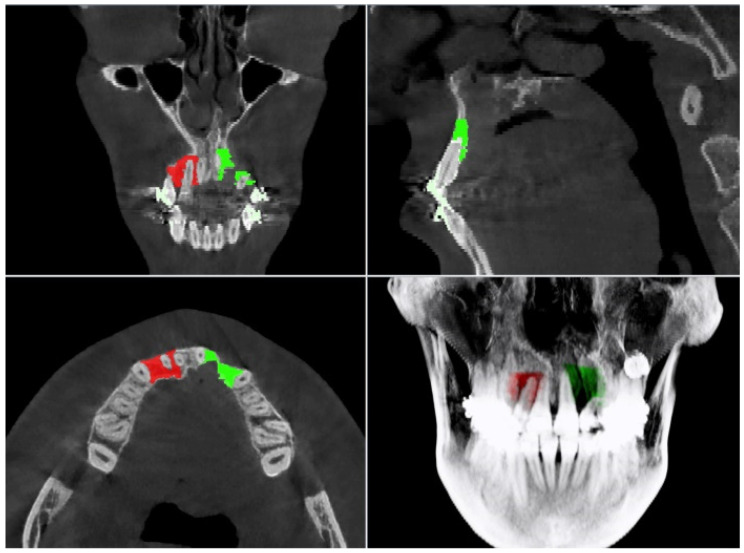
Multiplanar reconstructions with overlaid markings of cleft and non-cleft segmentation highlighted in green and red, respectively. The alveolar grafted bone area was traced on the horizontal computed tomography (CT) slices, using the drawing tools of VisNow software. The labial and palatal outline of bone localized between central incisor and canine were followed, excluding root of the lateral incisor if present.

**Figure 2 jcm-10-03524-f002:**
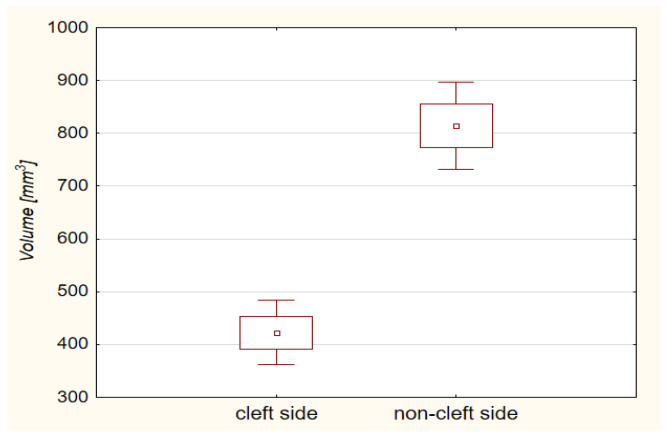
Box and whisker plot presenting the means, means ± standard deviations and means ± 1.96 × standard deviations of the cleft-side alveolar bone volume and the non-cleft side alveolar bone volume. The difference between the volumes was statistically significant (*p* < 0.0001, Student’s *t*-test).

**Figure 3 jcm-10-03524-f003:**
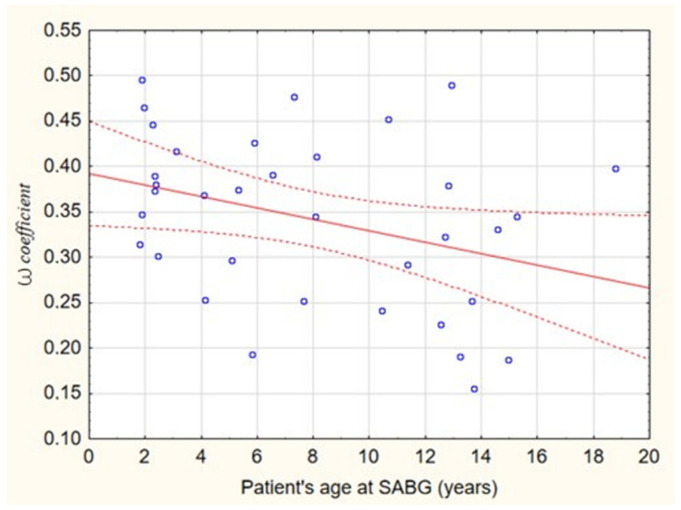
Scatter plot depicting the correlation between *Ꞷ* coefficient and patient’s age at SABG (*r* = −0.34, *p* = 0.045). Regression equation is given by: *Ꞷ* = 0.392 − 0.006 × SABG.

**Table 1 jcm-10-03524-t001:** Demographic and clinical characteristics of the study group. IQR-interquartile range. SABG—secondary alveolar bone grafting.

	All (*n* = 35)		
	Median	Range	IQR
Age at primary cleft repair (months)	8.1	5.0–20.6	1.5
Age at alveolar bone grafting (years)	8.4	1.8–18.8	8.3
Age at CT scan (years)	17.2	14.2–22.1	2.8
Time from SABG to CT scan (years)	9.9	1.8–17.2	8.0
Proportion of left to right clefts (%)	22/13 (62.9/37.1)		
Proportion of boys to girls (%)	17/18 (48.6/51.4)		

SABG: secondary alveolar bone grafting.

**Table 2 jcm-10-03524-t002:** Pearson’s correlation univariate analysis results between *Ꞷ* coefficient, age at cleft repair, age at SABG, and age at CT.

		*Ꞷ* Coefficient	Age at Cleft Repair (Months)	Age at SABG (Years)	Age at CT (Years)
Pearson’s correlation (*p* value)	*Ꞷ* coefficient	1			
	Age at cleft repair (months)	−0.24 (0.16)	1		
	Age at SABG (years)	−0.34 (0.045)	0.03 (0.432)	1	
	Age at CT (years)	0.29 (0.08)	0.20 (0.129)	0.20 (0.069)	1

**Table 3 jcm-10-03524-t003:** Multiple regression model with *Ꞷ* coefficient as dependent variable and age at SABG, age at CT as independent variables.

Dependent Variable	Independent Variables	Coefficient	Standard Error	*p* Value	R^2^ Partial	R^2^ of the Model	*p* Value
*Ꞷ* coefficient	Age at cleft repair (months)	−0.018	0.010	0.065	0.044	0.228	0.053
	Age at SABG (years)	−0.008	0.003	0.003	0.187		
	Age at CT (years)	0.011	0.008	0.186	0.215		

## Data Availability

Up on request from the corresponding author.

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
