# Peer review of "Alveolar Volume Following Different Timings of Secondary Bone Grafting in Patients with Unilateral Cleft Lip and Palate. A Pilot Study"

_jcm, 2021, doi:10.3390/jcm10163524_

Round 1

Reviewer 1 Report

Brief summary of review report

The manuscript has been well revised. However, I strongly believe that the authors performed a large number of SABG cases in early stage, they should not publish a retrospective study on a small number of cases, even if it is a “pilot study”. At least, they should conduct a power analysis and describe the estimation for how many cases they need to collect in the future study. If the aim is to overturn the conventional treatment policy of SABG in the period of canine eruption, I believe that a more careful research attitude is required.

Broad comments

The manuscript has been well revised. I strongly advise the authors to increase the number of participants or perform a power analysis.

Specific comments

  1. The descriptions of CT exposure parameters were changed. Although the voxel size is described as 0.3 x 0.3 mm – is it correct? The authors wrote it as 0.3 x 0.3 x 0.9 mm in the last version. If the acquired data was not isotropic volume data, the last description is correct. I would like to retract my previous comment. (L. 113)

Thank you for the revision of the relative coefficient. But uppercase and lowercase letters, and alphabet and Greek letter are confused in the current manuscript. (Formula 1, vertical axis in the Figure 3, and L.242)

Author Response

X

Dear Editor,

Thank you for allowing a resubmission of our manuscript, with an opportunity to address the reviewers’ comments. We are uploading our response to the reviewers’ comments and an updated manuscript.

Reviewer concerns:
Reviewer #1
Brief summary of review report
The manuscript has been well revised. However, I strongly believe that the authors performed a large number of SABG cases in early stage, they should not publish a retrospective study on a small number of cases, even if it is a “pilot study”. At least, they should conduct a power analysis and describe the estimation for how many cases they need to collect in the future study. If the aim is to overturn the conventional treatment policy of SABG in the period of canine eruption, I believe that a more careful research attitude is required.
Broad comments
The manuscript has been well revised. I strongly advise the authors to increase the number of participants or perform a power analysis.
Author response: According to the Reviewer’s suggestion the results power analysis were added to the manuscript. “The power analysis revealed that sufficient number of cases required for further prospective research should be 51 at least. The sample size should be increased by the potential dropout of patients to follow-up during clinical trial. If we assume that the 20% patient’s lost to follow-up, than 62 patients in total are required for the trial”
Specific comments
    The descriptions of CT exposure parameters were changed. Although the voxel size is described as 0.3 x 0.3 mm – is it correct? The authors wrote it as 0.3 x 0.3 x 0.9 mm in the last version. If the acquired data was not isotropic volume data, the last description is correct. I would like to retract my previous comment. (L. 113)
Author response: The voxel size was anisotropic, therefore, the previous version was restored. 
Thank you for the revision of the relative coefficient. But uppercase and lowercase letters, and alphabet and Greek letter are confused in the current manuscript. (Formula 1, vertical axis in the Figure 3, and L.242)
Author response: The manuscript was changed according to the Reviewer’s suggestion

Reviewer 2 Report

Authors investigated the timing of second alveolar bone grafting on UCLP through a retrospective study. I think your setting of relative coefficient compared non-cleft side is interesting and a good method. Your conclusion was introduced that younger is better for SABG. However, this is nothing new one because post report has described SABG at the older age has less bone formation on alveolar cleft. In general, this operation is performed at the age of 5-6 years or 9-10 years, whose time is before eruption of lateral incisor or canine respectively. So, the important thing is a strategy of SABG clinically. Authors should describe it and explain why the age was uniform in this study. I wondered about various age.

And I want to know why authors used age of cleft lip repair as an independent variable. I think the age of cleft lip repair has no relation with SABG. 

Author Response

Reviewer concerns:

Reviewer #2

Authors investigated the timing of second alveolar bone grafting on UCLP through a retrospective study. I think your setting of relative coefficient compared non-cleft side is interesting and a good method. Your conclusion was introduced that younger is better for SABG. However, this is nothing new one because post report has described SABG at the older age has less bone formation on alveolar cleft. In general, this operation is performed at the age of 5-6 years or 9-10 years, whose time is before eruption of lateral incisor or canine respectively. So, the important thing is a strategy of SABG clinically. Authors should describe it and explain why the age was uniform in this study. I wondered about various age.

Author response: Our conclusion is as follows: “we cautiously conclude that our research hypothesis was partially confirmed - alveolar bone volume on the cleft side seemed not to be less when SABG was performed in younger patients in comparison to SABG performed in older patients”. We would like to stress that we have not stated that “younger is better for SABG” but rather “younger seems not to be worse”.

The reviewer has mentioned in the comment that the age at SABG was uniform. In fact the age at SABG was not uniform at all. Table 1 demonstrates it:

                                                           Median             Range               Inter-Quartile Range

Age at alveolar bone grafting (years)     8,4            1.8-18.8                8.3

The reason for a wide age range in the current study was that not all patients received SABG at age 2-4 years (SABG performed between 2 and 4 yrs is the part of the protocol of treatment of UCLP at our institution). Some of them had SABG later, sometimes very late (the oldest age at SABG was 18.8 yrs). We selected our sample based primarily on the availability of CT images and it resulted in age variability

Comment:

And I want to know why authors used age of cleft lip repair as an independent variable. I think the age of cleft lip repair has no relation with SABG.

Author response: We observed a tendency to close the cleft lip and palate earlier and earlier in our institution (please mind that we do one-stage closure of the entire cleft in our institution). For example, patients with UCLP operated in years 1993-1998 had one-stage closure of the cleft performed at 9.2 months, while those operated in years 1999-2003 had one-stage closure of the cleft performed at 7.7 months (1). The age at cleft closure was associated with the outcome (2). Because of these 2 reasons we included age at cleft repair (not age at lip repair as suggested by the reviewer  but age at lip and palate repair) in the regression model as independent variable.

  1. Brudnicki A, Sawicka E, Fudalej PS. Maxillofacial morphology in post-pubertal patients with unilateral cleft lip and palate following early vs. late secondary alveolar bone grafting. J Craniomaxillofac Surg. 2021 Apr 24:S1010-5182(21)00121-9. doi: 10.1016/j.jcms.2021.04.012. Epub ahead of print. PMID: 33965325.
  2. Siegenthaler M, Bettelini L, Brudnicki A, Rachwalski M, Fudalej PS. Early versus late alveolar bone grafting in unilateral cleft lip and palate: Dental arch relationships in pre-adolescent patients. J Craniomaxillofac Surg. 2018 Dec;46(12):2052-2057. doi: 10.1016/j.jcms.2018.09.031. Epub 2018 Sep 29. PMID: 30416034.

Round 2

Reviewer 1 Report

1. Alphabet and Greek letters, italic and plain styles are still confused in the current revision (L.23 in the abstract, L.180, the vertical axis in Figure 3, L.195, L.203, L.206, and L.216).

2. The variable "surgeon performing SABG" remains in the description of the result of the multiple regression model and legend of Table 3. Is it OK not to delete it? (L.202, 216-217)

3. The p-value of "Age at SABG" in the multiple regression model is described as 0.027 in the text, although it is written as 0.003 in Table 3. In addition, the description of the regression coefficient confuses 0.0076 and 0.0078. Please check the corrections associated with the changes made to the model-- e.g. The R2 of the model should be written as 0.23 (or 0.228).

Author Response

Author response: All Reviewer’s suggestions (1-3) were applied in the text. Some confusions were caused due to “track changes” option in MS Word, therefore, it was switched off in the final version of the manuscript.

This manuscript is a resubmission of an earlier submission. The following is a list of the peer review reports and author responses from that submission.

Round 1

Reviewer 1 Report

This study aims to evaluate the impact of age on alveolar cleft bone grafting outcomes. Despite the fact the study is well-conducted, it presents some weak points.

The authors concluded that alveolar cleft bone grafting realized at an early age gets better results, but they did not mention any result about maxillary growth. They realized CT-scan at a mean age of 17.2 years, so after puberty, but no result about the amount of retro/hypomaxilly is precised.

Moreover, the results of the graft could depend on the tooth environment. The tooth going through the bone graft could improve and stabilize the result. Evaluating a bone graft result without any knowledge about tooth environment is a major bias.

Then, the authors emphasized the fact this is the only study comparing very early alveolar bone grafting (before 5) and the ones performed at older ages. It does not precise if alveolar cleft bone grafting performed between 2 and 4 years gets better results than ones performed between 5 and 6, which could also be considered as very early bone grafting. At 5 years, the child usually benefited from dentofacial orthopedics to increase transversal dimension of the maxilla and to restore lateral incisor space, which impacts alveolar bone grafting results, as well as it could be a major point for maxillary growth. Before the age of 4 years, dentofacial orthopedics is usually quite difficult to perform and so could not frame the operating action of bone grafting.

Finally, the series is very small for such a range of ages.

These different points should be added, precised and/or discussed in this article. Conclusions should be weighted considering all these factors and biases.

Reviewer 2 Report

The manuscript is well-organized. The authors hypothesized that SABG at a younger age (2 – 5 years) may improve the bone bridge in the alveolar cleft. I think this is a very interesting and highly unique point of view because there have been only a limited number of studies evaluating the effects of SABG in this age group.

However, the number of participants is decisively small. Especially, in multiple regression analysis, 15 cases are needed for one explanatory variable. The reviewer strongly recommends the authors collect a sufficient number of cases before conducting the analysis.

Brief summary of review report

The authors evaluated the impact of the timing of secondary bone grafting (SABG) in patients with unilateral cleft lip and palate in a wide age range (1.8 – 18.8 years). Using CT images of 35patients, the relative coefficient (W, the ratio of the acquired alveolar bone volume in cleft-side to that in non-cleft side) has been assessed statistically. In univariate analysis, the patient’s age at the SABG showed a weak negative relationship. The multiple regression analysis revealed that the model with variables authors chose was not significant statistically. They concluded that the alveolar bone volume on the cleft side tended to be enlarged in younger age.

Broad comments

The manuscript is well-organized. The authors hypothesized that SABG at the younger age (2 – 5 years) may improve the bone bridge in alveolar cleft. I think this is a very interesting and highly unique point of view because there have been only a limited number of studies evaluating the effects of SABG in this age group.

However, the number of participants is decisively small. Especially, in multiple regression analysis, 15 cases are needed for one explanatory variable. The reviewer strongly recommends the authors to collect a sufficient number of cases before conducting the analysis.

Specific comments

  1. Was patient with Simonarts’ band included or excluded? (L. 80-81)
  2. The reviewer thinks lamp voltage is not common expression. Tube voltage should be appropriate. Additionally, since voxel size should be a cube, it would be incorrect to have different lengths for the three sides. Please confirm the scanning parameters. (L. 95-98)
  3. The relative coefficient (W) may be confused with Kendall’s W. The reviewer recommends using another letter. (L. 129-131)
  4. Please describe the definition of Vcleft-side and Vnon-cleft-side. (Formula 1)
  5. Although the measurement adaptation was precisely indicated in section 2.5.. please indicate the number of raters and the interval between measurement.
  6. The expression “The correlation coefficient was also performed between…” is unnatural. It is not “performed” item. (L.138)
  7. An explanatory variable “surgeon” appeared in multiple regression analysis was not estimated in univariate analysis. Why was this variable added in multiple regression? (L.142-143)
  8. Since the subject’s age at alveolar bone grafting did not seem to be Gaussian distribution, the reviewer recommends using the median and IQR instead of the mean and standard deviation.
  9. Please indicate the results of every univariate analysis in table.
  10. In multiple regression analysis, 15 cases are needed for one explanatory variable. The number of subjects is decisively small. Additionally, it is an incorrect statement in statistics to express “tendency” when p-value close to the significance level. The regression coefficient = -0.0076 is too small comparing to the W coefficient that appear to be distributed from 0.15 to 0.50 -- is it clinically meaningful size? (L.178-184)
  11. Please mention about the disadvantage especially in donor site to perform the SABG at earlier age. Additionally, the age range of “2 – 5 years” is arbitrary; please explain why it is 5 instead of 4 or 6. (L.200)
  12. The (semi-)automatization in the tracing of alveolar bone bridging or defect has been reported. (L.245-247)